# Are Large Brainwave Foundation Models Capable Yet? Insights from Fine-tuning

**Na Lee** [* 1 2]  **Konstantinos Barmpas** [* 1 2 3]  **Yannis Panagakis** [2 3 4]  **Dimitrios Adamos** [1 2]  **Nikolaos Laskaris** [2 5]
**Stefanos Zafeiriou** [1 2]

## Abstract

Foundation Models have demonstrated significant success across various domains in Artificial Intelligence (AI), yet their capabilities for brainwave modeling remain unclear. In this paper, we comprehensively evaluate current Large Brainwave Foundation Models (LBMs) through systematic fine-tuning experiments across multiple Brain-Computer Interface (BCI) benchmark tasks, including memory tasks and sleep stage classification. Our extensive analysis shows that state-of-the-art LBMs achieve only marginal improvements (0.9%-1.2%) over traditional deep architectures while requiring significantly more parameters (millions vs thousands), raising important questions about their efficiency and applicability in BCI contexts. Moreover, through detailed ablation studies and Low-Rank Adaptation (LoRA), we significantly reduce trainable parameters without performance degradation, while demonstrating that architectural and training inefficiencies limit LBMs' current capabilities. Our experiments span both full model fine-tuning and parameter-efficient adaptation techniques, providing insights into optimal training strategies for BCI applications. We pioneer the application of LoRA to LBMs, revealing that performance benefits generally emerge when adapting multiple neural network components simultaneously. These findings highlight the critical need for domain-specific development strategies to advance LBMs, suggesting that current architectures may require redesign to fully leverage the potential of foundation models in brainwave analysis.

*Equal contribution  [1]Imperial College London  [2]Cogitat  [3]Archimedes / Athena Research Unit  [4]National and Kapodistrian University of Athens  [5]Aristotle University of Thessaloniki. Correspondence to: Na Lee <na.lee12@ic.ac.uk>.

*Proceedings of the $42^{nd}$ International Conference on Machine Learning*, Vancouver, Canada. PMLR 267, 2025. Copyright 2025 by the author(s).

## 1. Introduction

Brain-Computer Interface (BCI) technology promises a new way to interact with machines by creating direct communication between the human brain and computers. This technology is based on the analysis of brainwaves from electroencephalogram (EEG) recordings using advanced signal processing and, more recently, machine learning techniques. BCI technology finds application in various areas like emotion recognition [(Torres et al., 2020), (Xu et al., 2018)], epileptic seizure detection [(Alkawadri, 2019), (Djoufack Nkengfack et al., 2021)], robotic control (Irimia et al., 2012) and video gaming (Kerous et al., 2018). BCIs can also augment human abilities and have the potential to transform how we interact with our environment and each other, offering hope to people with disabilities to regain lost functions [(Chaudhary et al., 2016), (Luu et al., 2017), (Biasiucci et al., 2018), (Kumarasinghe et al., 2021), (Sharma et al., 2016)].

The early days of the BCI era placed human experts at the center of brainwave analysis, with manual feature extraction by neuroengineers regarded as the gold standard for many years [(Bashashati et al., 2007), (Handy, 2009), (Rao, 2013), (Nam et al., 2018), (McFarland et al., 2006)]. However, these hand-crafted features often fail to generalize effectively to real-world data, limiting their practicality in everyday BCI applications.

The advent of deep learning has made the need for manual feature extraction redundant, as new data-driven approaches led to state-of-the-art performance in various BCI paradigms [(Lawhern et al., 2018), (Santamaría-Vázquez et al., 2020), (Song et al., 2023), (Barmpas et al., 2023), (Bakas et al., 2022), (Wei et al., 2022)]. Although deep learning models have demonstrated impressive results, they generally demand substantial supervision and task-specific data collection, making the process both time-intensive and resource-demanding.

Foundation Models have recently emerged as a promising approach to address these limitations, showing remarkable results in various domains, particularly in Natural Language Processing and Computer Vision [(Brown et al., 2020), (Tou-

vron et al., 2023), (Mizrahi et al., 2023), (Paraperas Papantoniou et al., 2024)]. Inspired by this tremendous progress, researchers have begun to develop similar Large Brainwave Models (LBMs) to the domain of BCIs [(Jiang et al., 2024), (Cui et al., 2024), (Wang et al., 2025), (Jiang et al., 2025)]. Theoretically, these large models offer several potential advantages: they are capable of identifying complex patterns and relationships in EEG data, thanks to their extensive self-supervised pre-training on a wide array of unlabeled datasets. Thus, they demonstrate improved generalization, reducing the need for task-specific data collection and model training and create more robust and versatile BCI systems capable of adapting to various users, tasks and environments. In addition, their generative nature equips these models with a strong adaptability to novel downstream tasks while also enabling them to generate high-quality synthetic data, thus offering promising solutions for predicting brain activity and reconstructing corrupted brain signals (Barmpas et al., 2024a).

In this work, we explore the current state of Large Brainwave Models (LBMs) for EEG-based BCIs, adopting a structured multi-step methodology:

1. We begin by evaluating the performance of publicly available pre-trained state-of-the-art LBMs, specifically LaBraM (Jiang et al., 2024) and NeuroGPT (Cui et al., 2024), against traditional deep learning models. This comparison aims to highlight the advantages and limitations of LBMs in comparison to well-established techniques.

2. We investigate the application of Low-Rank Adaptation (LoRA) (Hu et al., 2021), a widely used method for parameter efficient fine-tuning (PEFT) of large pre-trained models across diverse tasks. Similarly to the exploration of time-series foundation models (Gupta et al., 2024), through ablation studies and extensive experimental analysis, we examine the efficacy of LoRA when applied to pre-trained LBMs.

3. We demonstrate that, by carefully selecting LoRA parameters, it is possible to significantly reduce the number of trainable parameters in pre-trained LBMs. Notably, this reduction is achieved without compromising model performance, offering a practical path towards more resource-efficient applications of LBMs in BCI systems.

## 2. Background

In recent years, several Large Brainwave Models (LBMs) have been introduced that promise strong generalization capabilities across various BCI paradigms. In this work, we will focus mainly on two of these LBMs, namely LaBraM (Jiang et al., 2024) and NeuroGPT (Cui et al., 2024).

### 2.1. LaBraM

LaBraM (Jiang et al., 2024) is a unified EEG foundation model designed to enable cross-dataset learning by segmenting EEG signals into channel-specific patches. Inspired by VQ-GAN (Esser et al., 2020), it employs vector-quantized neural spectrum prediction to train a semantically rich neural tokenizer, which encodes continuous raw EEG channel patches into compact neural codes, known as a neural codebook.

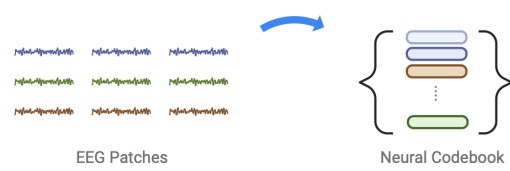

EEG Patches      Neural Codebook

*Figure 1.* Illustration of LaBraM's Neural Codebook

This neural codebook serves as a strong base for pre-training the foundation model. LaBraM follows a two-step pre-training approach: first training of the neural codebook with target objective being the reconstruction of the fourier amplitude and phase of the EEG patch. Then, the core training of the foundation model, where the model learns by predicting the original neural codebook for masked EEG channel patches.

LaBraM was pre-trained on approximately 2,500 hours of diverse EEG signals sourced from around 20 datasets. Its effectiveness was validated on a variety of downstream tasks, demonstrating its versatility and robustness for different EEG-based applications.

### 2.2. NeuroGPT

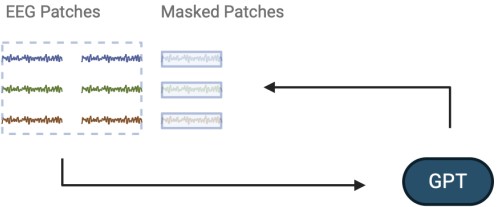

EEG Patches      Masked Patches      GPT

*Figure 2.* Illustration of NeuroGPT's Auto-Regressive Training

NeuroGPT (Cui et al., 2024) is a foundation model that combines an EEG encoder with a GPT-based architecture. The EEG encoder draws inspiration from the widely recog-

nized deep learning framework EEGConformer (Song et al., 2023) that utilizes a spatio-temporal convolutional feature extraction followed by a series of self-attention layers. The model leverages GPT-style self-supervised training, employing an auto-regressive approach (Brown et al., 2020) where it predicts the next masked token based on preceding tokens. This training paradigm enables NeuroGPT to capture complex temporal and spatial patterns in EEG data, making it a robust foundation model for a variety of downstream EEG-based applications.

NeuroGPT is pre-trained on recordings from the Temple University Hospital (TUH) EEG Corpus (Obeid & Picone, 2016), a comprehensive dataset that offers diverse and extensive data for model training. Specifically, NeuroGPT was trained on 20,000 EEG recordings from the TUH corpus dataset with a total duration of 5656 hours.

### 2.3. Low-Rank Adaptation (LoRA)

Low-Rank Adaptation is a PEFT technique that introduces low-rank updates to pre-trained models, significantly reducing the number of trainable parameters. In standard fine-tuning, the full weight matrix $\mathbf{W} \in \mathbb{R}^{d \times k}$ is updated during training which can be computationally expensive for large-scale models. LoRA, instead, decomposes the update into the product of two low-rank matrices. Mathematically:

$$\Delta\mathbf{W} = \mathbf{AB},$$

where $\mathbf{A} \in \mathbb{R}^{d \times r}$ and $\mathbf{B} \in \mathbb{R}^{r \times k}$, with $r \ll \min(d, k)$. During fine-tuning, the original weight matrix $\mathbf{W}$ remains frozen, and only the low-rank matrices $\mathbf{A}$ and $\mathbf{B}$ are trained. The effective weight becomes:

$$\mathbf{W}^{'} = \mathbf{W} + \Delta\mathbf{W} = \mathbf{W} + \mathbf{AB}.$$

This approach greatly reduces the number of trainable parameters from $d \times k$ to $r \times (d + k)$, where $r$ is the rank of the decomposition. The low-rank matrices capture task-specific adaptations while preserving the original model's pre-trained knowledge. For example, in transformer-based layers, LoRA is typically applied to the weight matrices of key, query or value projections in self-attention layers, significantly reducing computational and memory demands. Mathematically, during inference, the computational overhead of LoRA is negligible because the low-rank updates $\Delta\mathbf{W}$ are pre-computed. This constitutes LoRA an effective method for adapting large pre-trained models to specific tasks, where fine-tuning efficiency and generalization are critical.

## 3. Analysis

### 3.1. Data Preprocessing

All models were evaluated in downstream classification tasks for the following five benchmark EEG datasets (Lee et al., 2025): Motor paradigm in High Gamma (Schirrmeister et al., 2017), the ERP (Event-Related Potential) paradigm from Korean University (Hong-Kyung et al., 2019), a Working Memory dataset (Pavlov et al., 2022), Physionet's sleep staging dataset, Sleep-EDF (Kemp et al., 2000) and Eyes Open vs Closed classification on the Physionet Motor dataset (Schalk et al., 2004). These tasks were selected to capture a diverse range of BCI paradigms and the datasets were specifically chosen for their minimal spurious artifacts, reducing the likelihood of specious performance during training (Lee et al., 2025).

For each of the baseline Large Brainwave Models, benchmark data was preprocessed to match the input data structure used during pre-training:

1. For LaBraM, a sample frequency of 200Hz was used, a bandpass filter from 0.5-45Hz was applied, as well as notch filters at 50Hz, 60Hz and 100Hz to remove powerline noise. Trials were cut into 1s patches taken across channels, to give 256 patches per sample. In addition to the EEG trial data, temporal and spatial embeddings for each sample were given as input to the model. Temporal embeddings include each patch's temporal position within the length of the trial, whereas spatial embeddings include the position of the patch's channel within a global list of all known electrodes. Only data from electrodes which were present in the global list provided were used.

2. For the NeuroGPT model, the data were resampled to 250Hz and a bandpass filter of 0.05-100Hz was applied. Similarly to LaBraM, notch filters at 50Hz, 60Hz and harmonics were also applied. NeuroGPT's input data need to include a specific set of channels in a fixed order. Therefore, for each benchmark dataset, we only use data from electrodes that are present in the pre-training data. For any expected channels which are not included in the benchmark data, the nearest available electrode's data is used (if the location is within a few centimeters), otherwise the channel data are set to zero.

For all downstream datasets, Common Average Re-referencing (CAR) was applied across all channels to reduce noise.

*Table 1.* Classification accuracy of finetuned foundation models and standard deep learning architectures, reported as mean (std). Each trained/finetuned for 20 epochs with 10 fold cross-validation. Trainable parameters include the size of the classification heads. Bold values indicate best performance per task or overall.

| MODEL | MOTOR | ERP | MEMORY | SLEEP | EYES | MEAN | PARAMS |
|---|---|---|---|---|---|---|---|
| EEGNET | 0.657 (.087) | **0.912** (.009) | 0.660 (.022) | 0.624 (.037) | 0.803 (.061) | 0.731 (.024) | 2,394 |
| EEG-INCEPTION | 0.590 (.087) | 0.896 (.007) | **0.669** (.021) | 0.688 (.057) | 0.823 (.038) | 0.733 (.021) | 22,366 |
| LABRAM | 0.614 (.096) | 0.911 (.013) | 0.643 (.040) | **0.704** (.025) | 0.840 (.041) | 0.742 (.023) | 5,854,288 |
| NEUROGPT (FULL MODEL) | 0.682 (.083) | 0.904 (.012) | 0.610 (.052) | 0.665 (.030) | 0.821 (.052) | 0.736 (.025) | 78,536,146 |
| NEUROGPT (ENCODER) | **0.695** (.085) | 0.908 (.012) | 0.634 (.035) | 0.647 (.024) | **0.843** (.045) | **0.745** (.027) | 717,958 |

*Table 2.* P-values of paired-t tests between EEGInception and finetuned foundation models. Bold values indicate statistically significant result ($p < 0.05$)

| MODELS | MOTOR | ERP | MEMORY | SLEEP | EYES |
|---|---|---|---|---|---|
| EEGINCEPTION / LABRAM | 0.5860 | **0.0123** | 0.1090 | 0.2995 | 0.4468 |
| EEGINCEPTION / NEUROGPT (FULL MODEL) | **0.0314** | **0.0401** | **0.0041** | 0.2226 | 0.8979 |
| EEGINCEPTION / NEUROGPT (ENCODER) | **0.0072** | **0.0051** | **0.0399** | **0.0495** | 0.1056 |

## 3.2. Comparing Brainwave Foundation Models with Deep Learning Models

The performance of large foundation models varies dramatically between domains. In some areas, large-scale pre-training has revolutionized task performance by enabling the models to generalize across a broad range of applications, often surpassing traditional methods. However, in other domains, foundation models sometimes fail to demonstrate significant advantages and are even outperformed by simpler, domain-specific baselines. It is therefore important to critically measure the effectiveness of recent Large Brainwave Foundation Models (LBMs). To achieve this, here we systematically evaluate the performance of fine-tuned Large Brainwave Foundation Models in comparison to other deep learning architectures. By benchmarking on a variety of tasks and datasets as described in section 3.1, we aim to assess whether fine-tuned LBMs consistently provide an advantage over more specialized or traditional approaches.

We perform finetuning on three configurations: the pre-trained LaBraM base model, the pre-trained NeuroGPT model and the encoder-only module of the pre-trained NeuroGPT model (as discussed in (Cui et al., 2024) where the authors claim that fine-tuning the encoder alone produces similar or improved results over the full model). Each configuration was trained for 20 epochs (to avoid overfitting) and evaluated using 10-fold subject-independent cross-validation, where samples were split on a subject level such

that no participant would be present in both the training and validation sets. To perform the downstream tasks, untrained classification heads were added to the pre-trained transformer-based Large Brainwave Foundation Models before finetuning. The size and structure of the classifier depend on the latent dimension of the model and the number of target classes for the given benchmark. The exact architecture of the classifiers is given in Table 4:

1. For LaBraM, a simple dropout and fully connected layer were used

2. For NeuroGPT, a three-layer MLP with dropout and ELU activations were used

Using the same fine-tuning setup, we performed a similar training (from scratch) process for the EEGNet and EEGInception models to provide a basis for comparison. As shown in Table 1, standard deep learning baselines can achieve comparable or even superior performance to some large models. However, NeuroGPT outperforms all other models on average, including both standard deep learning baselines and other large foundation models. While NeuroGPT might not lead in every benchmark task, it consistently demonstrates strong average performance.

Both foundation models (LaBraM and Neuro-GPT) have in their pre-training datasets paradigms that include motor-, ERP-, sleep- and eyes-related tasks. The results in Table

*Table 3.* Classification accuracy of foundation models where all parameters except classification heads are frozen. Each trained for 20 epochs with 10 fold cross-validation. Bold values indicate best performance per task or overall.

| MODEL | MOTOR | ERP | MEMORY | SLEEP | EYES | MEAN ACCURACY |
|---|---|---|---|---|---|---|
| LABRAM | 0.297 | **0.884** | **0.670** | **0.608** | 0.717 | 0.635 |
| NEUROGPT (FULL MODEL) | 0.366 | **0.884** | 0.656 | 0.597 | 0.734 | 0.647 |
| NEUROGPT (ENCODER) | **0.431** | 0.883 | 0.655 | 0.602 | **0.746** | **0.663** |

*Table 4.* Classification heads for each model configuration where n_cls is the number of classes for each benchmark task

| MODEL | CLASSIFIER |
|---|---|
| LABRAM | Linear (200, n_cls), Dropout (0.5) |
| NEUROGPT FULL MODEL | Linear (1024,256), ELU, Dropout (0.5), Linear (256,32), ELU, Dropout (0.3), Linear (32, n_cls) |
| NEUROGPT ENCODER | Linear (2160,256), ELU, Dropout (0.5), Linear (256,32), ELU, Linear (32, n_cls) |

1 show that foundation models achieve comparable performance to baseline models in ERP, sleep and eyes tasks. NeuroGPT significantly outperforms baseline models in motor. Interestingly, in the memory task—which was not explicitly included in the pre-training datasets—baseline models slightly outperform foundation models. Although the performance margin is 1.2% compared to the next-best baseline standard deep learning model, this indicates that, while these foundation models represent a promising step toward advancing Large Brainwave Foundation Models, their substantial benefits over traditional approaches have yet to be fully realized. In summary:

> Standard deep learning baselines trained only on specific tasks can achieve **comparable** performance to fine-tuned pre-trained large brainwave foundation models while having only a fraction of trainable parameters.

Testing the generalization capabilities of large foundation models is also crucial. Therefore, we performed the same fine-tuning process as in the above-mentioned tasks but kept the pre-trained foundation models frozen and trained only the classification heads during the fine-tuning step. From the results in Table 3, training just the classification heads yields models that lack behind traditional deep learning approaches by a large margin of almost 8-10%. This demonstrates the necessity of full-model fine-tuning, and in turn makes parameter efficient fine-tuning (PEFT) techniques like LoRA extremely valuable, which we will explore in the next sections.

### 3.3. Low-Rank Adaptation Exploration in Brainwave Foundation Models

In this section, we investigate how fine-tuning strategies, such as Low-Rank Adaptation, influence the performance of these pre-trained large brainwave foundation models, shedding light on ways to maximize the utility of these models across diverse domains. Specifically, we explore the performance and parameter efficiency of LoRA when applied in different ways to finetuning these pre-trained large brainwave foundation models.

In our analysis, we use the original formulation of LoRA wherein each target module's weight matrix $\mathbf{W}$ is adapted with two low-rank matrices $\mathbf{A}$, $\mathbf{B}$ with a chosen rank, $r$. We choose not to adapt any bias terms and leave them frozen during finetuning. When adapting attention modules we treat queries, keys and values as a single combined matrix $\mathbf{W}_{qkv}$, however the output projection is left frozen. Furthermore, we also apply LoRA to fully connected and convolutional layers in our analysis to effectively evaluate the importance of these layers during the finetuning process. In all experiments, the scaling factor $\alpha$ (as described in (Hu et al., 2021)) is set to 8.

#### 3.3.1. LOW-RANK ADAPTATION IN ALL LAYERS

In this subsection, we investigate the effect of LoRA when different ranks are applied to the attention and fully-connected layers of the large brainwave foundation models. The rank of the convolutional layers $r_c$ is set to the maximum power of 2 which would not lead to adapters with a greater number of parameters than the original weight matrices. For LaBraM $r_c = 4$, and NeuroGPT $r_c = 8$ for both configurations, the full model and encoder only. Given the fixed $r_c$, we experiment with different ranks for the attention and fully connected modules $r \in \{1, 2, 4, 8, 16\}$.

*Table 5.* Performance of foundation models finetuned using LoRA with varying ranks for attention and fully connected layers. Ranks for LoRA adapters on convolutional layers are fixed to the maximum possible for the given model. Bold values indicate best performance per task or overall.

| MODEL | RANK | ERP | MEMORY | SLEEP | EYES | MEAN ACCURACY | TRAINABLE PARAMETERS |
|---|---|---|---|---|---|---|---|
| | 1 | **0.905** | 0.624 | **0.729** | 0.839 | 0.774 | 34,149 |
| | 2 | 0.902 | **0.643** | 0.725 | 0.836 | **0.777** | 67,749 |
| LABRAM | 4 | 0.902 | 0.638 | 0.715 | 0.822 | 0.770 | 134,949 |
| | 8 | 0.901 | 0.636 | 0.712 | 0.827 | 0.769 | 269,349 |
| | 16 | 0.902 | 0.626 | 0.708 | **0.845** | 0.770 | 538,149 |
| | 1 | 0.884 | 0.650 | 0.643 | 0.788 | 0.741 | 373,218 |
| NEUROGPT | 2 | 0.885 | 0.650 | 0.635 | **0.800** | 0.743 | 467,226 |
| FULL | 4 | 0.884 | **0.656** | 0.643 | 0.796 | **0.745** | 655,242 |
| MODEL | 8 | **0.885** | 0.642 | 0.642 | 0.798 | 0.742 | 1,031,274 |
| | 16 | 0.884 | 0.646 | **0.643** | 0.791 | 0.741 | 1,783,338 |
| | 1 | 0.896 | 0.643 | 0.643 | 0.796 | 0.744 | 573,866 |
| NEUROGPT | 2 | 0.897 | 0.641 | 0.643 | 0.801 | 0.746 | 577,706 |
| ENCODER | 4 | **0.897** | 0.643 | **0.644** | 0.799 | 0.746 | 585,386 |
| | 8 | 0.897 | **0.643** | 0.643 | **0.806** | **0.747** | 600,746 |
| | 16 | 0.894 | 0.642 | **0.644** | 0.801 | 0.745 | 631,466 |

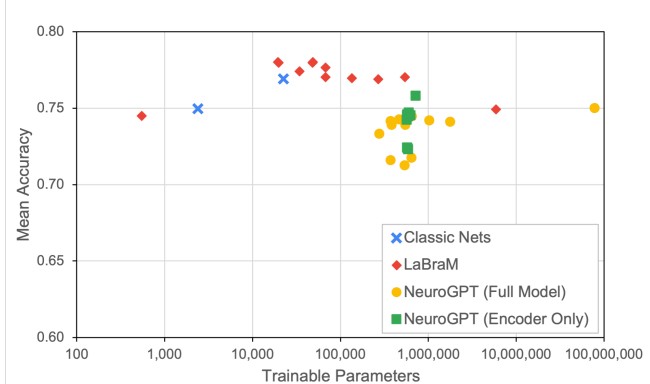

*Figure 3.* Number of trainable parameters against mean accuracy across all four downstream tasks.

As it is shown in Table 5:

> Using LoRA in large brainwave foundation models can significantly **reduce** the number of trainable parameters **without** compromising model performance

Theoretically, we would expect performance to increase with rank. For NeuroGPT, that assumption holds while for LaBraM its performance peaks at rank = 2.

### 3.3.2. LOW-RANK ADAPTATION - ABLATION STUDIES

In order to perform extensive experimental analysis to further understand the importance of each element of the large brainwave foundation model in the fine-tuned downstream

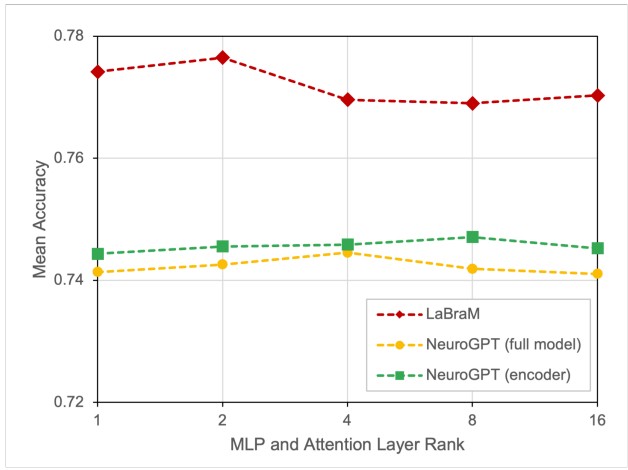

*Figure 4.* Mean accuracy vs rank of attention and fully connected layers, given fixed rank for convolutional layers

performance, we performed a series of ablation studies using the LoRA technique. For each of the three model configurations, we use the rank that produces the best average classification performance across all benchmarks, denoted as $r'$. We then apply LoRA to all possible combinations of convolution, attention and fully-connected layers.

As it is shown in Table 6:

1. Performing LoRA only on a specific layer, e.g. attention or convolution, usually yields lower performance compared to a combination of two or three of these layers.

*Table 6.* Performance of foundation models finetuned using LoRA adapters on different combinations of layer types. Ranks of attention and fully connected layers are fixed to the best performing rank $r'$ for each model configuration as given in Table 5. Bold values indicate best performance per task or overall.

| MODEL | LoRA LAYERS | KU ERP | MEMORY | SLEEP EDF | EYES OPEN/-CLOSED | MEAN ACCURACY | TRAINABLE PARAMETERS |
|---|---|---|---|---|---|---|---|
| LABRAM $r' = 2$ | ATTENTION | 0.902 | 0.644 | 0.722 | **0.852** | **0.780** | 19,401 |
| | FC | 0.900 | 0.657 | 0.727 | 0.835 | **0.780** | 48,201 |
| | CONV | 0.884 | **0.670** | 0.659 | 0.768 | 0.745 | 549 |
| | ATTENTION, FC | 0.899 | 0.623 | 0.717 | 0.843 | 0.770 | 67,401 |
| | ATTENTION, CONV | **0.904** | 0.652 | 0.729 | 0.834 | **0.780** | 19,749 |
| | FC, CONV | 0.901 | 0.659 | **0.732** | 0.828 | **0.780** | 48,549 |
| NEUROGPT FULL MODEL $r' = 4$ | ATTENTION | 0.883 | 0.656 | 0.599 | 0.726 | 0.716 | 374,754 |
| | FC | 0.884 | 0.656 | 0.579 | 0.732 | 0.713 | 542,658 |
| | CONV | 0.884 | **0.657** | 0.620 | 0.772 | 0.733 | 279,210 |
| | ATTENTION, FC | 0.883 | 0.656 | 0.594 | 0.736 | 0.717 | 646,722 |
| | ATTENTION, CONV | 0.882 | 0.654 | 0.635 | 0.785 | **0.739** | 383,274 |
| | FC, CONV | **0.884** | 0.646 | **0.637** | **0.788** | **0.739** | 551,178 |
| NEUROGPT ENCODER $r' = 8$ | ATTENTION | 0.883 | 0.652 | 0.612 | 0.750 | 0.724 | 573,026 |
| | FC | 0.883 | **0.655** | 0.607 | 0.751 | 0.724 | 580,706 |
| | CONV | 0.894 | 0.644 | 0.631 | 0.801 | 0.742 | 570,026 |
| | ATTENTION, FC | 0.883 | 0.647 | 0.613 | 0.749 | 0.723 | 592,226 |
| | ATTENTION, CONV | 0.896 | 0.639 | **0.643** | 0.800 | **0.745** | 581,546 |
| | FC, CONV | **0.897** | 0.644 | 0.635 | **0.803** | **0.745** | 589,226 |

2. The combination of convolution layers with either fully connected or attention layers has the best average performance across all benchmarks.

3. Interestingly, the combinations of attention and convolution and fully-connected and convolution demonstrate the same performance. This unveils that for these state-of-the-art brainwave foundation models the attention layers might not capture as important information as their temporal encoding parts.

Therefore, we can conlude that:

> LoRA in large brainwave foundation models can demonstrate performance benefits when used in **combination** of two or three different types of layers.

### 3.3.3. EFFECT OF DROPOUT ON LOW-RANK ADAPTATION

As demonstrated, foundation models can marginally outperform traditional deep learning models on average and, when finetuned with LoRA, yield an additional performance boost. To investigate this further, we aimed to dive deeper into the LoRA training process for the model with best average performance from Table 6 (LaBraM) and explore potential strategies for further enhancing its performance.

Therefore, in this subsection we explore the effects of introducing dropout in the parameter space of LoRA's low-rank matrices. To achieve this we apply LoRA to the LaBraM model, adapting attention, fully-connected and convolutional layers as in Table 5, with identical setup except this time introducing a dropout for each adapter. The findings of (Dettmers et al., 2023) suggest a dropout probability of 0.1 when applying LoRA to 7B and 13B parameter models, or 0.05 for 33B and 65B models. The size of the full LaBraM base model is only around 5.8M parameters, therefore we select a relatively high dropout of 0.5.

As it is shown in Table 7:

1. Introducing a dropout to LoRA adapters can match or improve classification performance.

2. Dropout's improvements in classification accuracy typically increase with rank.

3. Performance is more positively affected in the memory and sleep classification tasks.

## 4. Discussion

Foundation models have revolutionized numerous fields in computer science, enabling breakthroughs in natural language processing, computer vision and other domains. While early efforts have been made to develop Large Brainwave Foundation Models (LBMs), these models have yet to reach their full potential. In this work, we investigated

*Table 7.* LaBraM finetuned using LoRA with varying ranks for attention and fully connected layer adapters. Each value is the *difference* in classification accuracy between using a dropout with probability 0.5 vs no dropout.

| RANK | KU ERP | MEMORY | SLEEP EDF | EYES OPEN/CLOSED | MEAN |
|---|---|---|---|---|---|
| 1 | -0.002 | +0.046 | +0.007 | -0.005 | +0.011 |
| 2 | +0.003 | +0.025 | +0.010 | +0.005 | +0.011 |
| 4 | +0.004 | +0.032 | +0.019 | +0.012 | +0.017 |
| 8 | +0.006 | +0.034 | +0.024 | +0.012 | +0.019 |
| 16 | +0.006 | +0.042 | +0.029 | -0.014 | +0.016 |

fine-tuning techniques applied to two state-of-the-art Large Brainwave Foundation Models.

Our findings reveal that large pre-trained models which offer interpretability insights (LaBraM) outperform standard deep learning baselines and black-box models (NeuroGPT). However, the margin of improvement is small when considering the relative sizes of the models: for example, a fully fine-tuned LaBraM has over 2000 times more trainable parameters than EEGNet. This finding signifies the importance of developing more efficient large brainwave models and the need to develop new domain-specific training techniques to train these large models.

Additionally, we conducted an in-depth study of the widely used Low-Rank Adaptation technique. Our results demonstrate that applying LoRA to large brainwave foundation models can substantially reduce the number of trainable parameters without sacrificing performance. However, through a series of ablation studies, we uncovered that performance improvements with LoRA are achieved only when it is applied to a combination of two or three different types of layers. This raises important questions about the pre-training processes used to develop these large models, unveiling cross-stage dependencies (rather than being limited e.g to attention layers) and suggesting that their architecture and training methodologies may require refinement and domain-specific training techniques to better capture the underlying nature of brainwave signal.

Previous works in the field of causal reasoning for deep learning brainwave models (Barmpas et al., 2024b) as well as LBMs (Barmpas et al., 2024a) have showcased important training considerations that one must take into account when training LBMs. These can be used in conjunction with this work to further guide the research community in the development of efficient LBMs.

We believe the future of LBMs should go beyond merely adopting transfer techniques from other domains. Instead, they should integrate domain-specific knowledge—such as leveraging various EEG modalities—and employ tailored training strategies, like brain-inspired masking techniques. These are essential elements to fully capture the diverse nature of EEG and build an efficient and effective LBM that

will largely outperform all current state-of-the-art baselines in various tasks with minimum required fine-tuning.

## 5. Conclusion

In this work, we investigate fine-tuning techniques for large brainwave foundation models (LBMs), providing a comprehensive evaluation of their performance in a range of downstream BCI benchmark tasks. Our experiments show that, despite their scale and pre-training, current fine-tuned LBMs underperform compared to standard deep learning models, which have significantly fewer trainable parameters. Furthermore, we demonstrate that the Low-Rank Adaptation (LoRA) fine-tuning technique can effectively reduce the trainable parameters of LBMs without compromising performance, but usually when applied to a combination of two or three different types of layers.

To the best of our knowledge, this is the first study to objectively and systematically assess the fine-tuned performance of LBMs on a diverse and carefully curated set of BCI downstream tasks. Furthermore, similarly the study in time-series foundation models (Gupta et al., 2024), this work pioneers the use of LoRA in the context of brainwave foundation models, a field that remains largely unexplored. This extensive study highlights critical considerations for the research community, emphasizing the need for more efficient and effective approaches to developing and fine-tuning Large Brainwave Foundation Models (LBMs).

## Impact Statement

This paper presents work whose goal is to advance the field of Machine Learning. There are many potential societal consequences of our work, none which we feel must be specifically highlighted here.

## Acknowledgments

This work was supported by the EPSRC Turing AI Fellowship (Grant Ref: EP/Z534699/1): Generative Machine Learning Models for Data of Arbitrary Underlying Geometry (MAGAL).

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
